# Sorption–Biological Treatment of Coastal Substrates of the Barents Sea in Low Temperature Using the *Rhodococcus erythropolis* Strain HO-KS22

**DOI:** 10.3390/microorganisms13092181

**Published:** 2025-09-18

**Authors:** Vladimir Myazin, Maria Korneykova, Nadezhda Fokina, Ekaterina Semenova, Tamara Babich, Milana Murzaeva

**Affiliations:** 1Agrarian and Technological Institute, People’s Friendship University of Russia (RUDN University), 117198 Moscow, Russia; korneykova_mv@pfur.ru (M.K.); murzaeva_msh@pfur.ru (M.M.); 2Institute of North Industrial Ecology Problems-Subdivision of the Federal Research Centre “Kola Science Centre of Russian Academy of Science”, 184209 Apatity, Russia; nadezdavf@yandex.ru; 3Winogradsky Institute of Microbiology, Research Center of Biotechnology of the Russian Academy of Sciences, 119071 Moscow, Russia; mkatusha82@mail.ru (E.S.); microb101@yandex.ru (T.B.)

**Keywords:** Arctic zone, shorelines, oil pollution, bioremediation, *Rhodococcus erythropolis*, sorbents

## Abstract

The efficiency of the sorption–biological method for treatment of oil-polluted coastal substrates (soil and sand) of the Barents Sea under low temperature (10 °C) using the active hydrocarbon-oxidizing bacterial strain *Rhodococcus erythropolis* HO-KS22 was assessed in the laboratory. The highest rate of hydrocarbon degradation was in sand polluted with a low-density oil emulsion and in soil polluted with a medium-density oil emulsion. Sorption–biological treatment increased the rate of hydrocarbon degradation in sand by 3–4 times during the first month and enhanced the overall efficiency by 20% over a three-month period. The use of sorbents (granular activated carbon, thermally activated vermiculite and peat) both in sand and soil prevents secondary pollution of coastal ecosystems, since it significantly reduces the hydrocarbons’ desorption and their leaching by water. *Rhodococcus erythropolis* HO-KS22, in combination with sorbents, can be applied during the biological remediation of coastal sandy substrates following the initial removal of emergency oil spills. However, for biological treatment of oil-polluted soils of the Barents Sea coast, further selection of active strains of hydrocarbon-oxidizing bacteria resistant to low pH values and temperatures typical for this region is necessary. The use of microbiological preparations without taking into account the soil and climatic factors of the region may be ineffective, which will increase the cost of remediation of the territory without significantly improving its condition.

## 1. Introduction

The Arctic continental shelf contains significant reserves of hydrocarbons, which drives competition for the exploration of these resources. Hydrocarbon deposits on the shelves of the Barents and Kara Seas represent a strategic reserve of oil and gas in the long term. Their development may lead to rapid urbanization and industrialization of the region. The primary risk zone includes the coastal areas of the Arctic seas, particularly in connection with the expansion of the Northern Sea Route. The spilled oil may be deposited on shorelines, which may have long-term impacts on wildlife, coastal communities’ health, and socioeconomic conditions [1]. Climate change is also driving industrialization and increasing shipping activity, opening up great areas of the northern seas. Climate change is even more significant for Arctic regions, which are warming at nearly three times the global rate [2]. Expected increases in shipping across the Arctic may pose a danger of collisions and shipwrecks [3]. The rise of storm frequency with high waves and wind speeds in the Arctic should also be considered [4].

The Barents Sea is located beyond the Arctic Circle at high latitudes. At the same time, its southwestern part does not freeze in winter due to the influence of the North Atlantic Current. The coastline is heavily indented, with numerous capes, bays, coves, and fjords. The location of the Barents Sea at high latitudes and its connection with the Atlantic Ocean and the Central Arctic Basin determine the polar maritime climate, which is characterized by a long winter, a short cold summer, small annual amplitude of air temperature, and high relative humidity. At the same time, the large meridional extent of the sea, the influx of large masses of warm Atlantic waters in the southwest, and the influx of cold waters from the Arctic Basin create climatic differences from place to place.

The coastline of the Barents Sea is represented by rocky shores, sandy, gravelly, and stone beaches, peat, and meadow shores. The intensity of tidal processes forms extensive tidal shallows.

Severe climatic conditions and short food chains make the marine ecosystem of the Barents Sea particularly sensitive to chemical pollution, including oil. However, the waters and coast of the Barents Sea are impacted by high transport and industrial loads: the sea connects the port of Murmansk with the ports of eastern and western countries; the waters of the southeastern part of the sea are some of the most explored for hydrocarbon reserves on the Russian shelf. In addition, the Barents Sea is an area of intensive fishing.

Intensive development of the Arctic region has revealed its high vulnerability and low environmental sustainability under anthropogenic impact. The Arctic currently suffers chronically from human activities, including oil spills and heavy metal pollution. Published scientific reviews note an increase in the number of publications devoted to hydrocarbon removal specifically in cold regions [1,3,5,6,7,8,9]. The largest number of publications is devoted to the treatment of diesel fuel, while the Arctic territory is often subject to pollution by oil and other petroleum products, for example, lubricant oils used to ensure the operation of engines and other mechanisms.

Special attention should be given to the development of strategies for cleaning and remediating the coastline, as oil discharged onto the Arctic shore can persist for extended periods [10]. At present, most cleanup methods—mechanical, thermal, and chemical—are costly and not consistently effective [11], and the specific cleanup method will be greatly influenced by shoreline types and oil characteristics [12,13,14]. Additionally, shoreline cleanup in cold regions can result in different degrees of negative impacts [1]. Given that the time between a spill and the arrival of a response team on a remote Arctic coast may take weeks, the deposition of hydrocarbons on the coastline becomes inevitable. When an oil spill occurs at sea, a water–oil emulsion is released onto the shore, which differs in its properties from oil. This greatly reduces the effectiveness of traditional mechanical removal and containment techniques for polluted substrates.

Under these conditions, and considering the logistical challenges of implementing alternative recovery strategies for Arctic coastal areas, bioremediation emerges as one of the most cost-effective cleanup approaches [3,15]. It may include natural attenuation, biostimulation, and bioaugmentation [16]. Bioremediation methods have a low potential impact on most shoreline types and have the least adverse impact on the habitat [1].

Therefore, one of the possible solutions for treatment pollution is bioremediation, i.e., using the metabolic abilities of natural microbiota [17]. Even at low temperatures, some microorganisms can survive only on hydrocarbons [3,18,19]. The main process of changes in the composition, structure, and properties of hydrocarbons is biochemical degradation. The mechanism and kinetics of these reactions have been studied quite well [20,21,22,23]. The most accessible for biochemical oxidation are aliphatic hydrocarbons of normal structure, followed by isoalkanes, cycloalkanes, alkylated cycloalkanes, and arenes. During the transformation process, the elemental composition (reduction in the proportion of hydrocarbons and an increase in the proportion of O-, S-, Hal-containing compounds) and structure (redistribution of homological and isological series towards hybrid condensates of the asphaltene type) of hydrocarbons change.

A relatively large group of hydrocarbon-degrading microorganisms is present in soils. This group includes bacteria of the genera *Alcaligenes*, *Agrobacterium*, *Arthrobacter*, *Bacillus*, *Flavobacterium*, *Pseudomonas*, *Rhodococcus*, *Mycobacterium*, *Streptomyces*, and others; filamentous fungi of the genera *Aspergillus*, *Mortierella*, *Penicillium*, and *Trichoderma;* and yeasts such as *Aureobasidium*, *Candida*, and *Rhodotorula* [24,25,26]. Hydrocarbon-degrading microorganisms differ in their ability to degrade hydrocarbons of different classes.

Bioaugmentation is introducing hydrocarbon-degrading microorganisms into oiled shorelines for hydrocarbon biodegradation [27]. The primary challenge hindering the further development of biotechnologies for the remediation of polluted areas is the instability of microbial communities (both natural and synthetic) under the stress conditions of low temperatures and high salinity typical of the marine environment. In the north, this may become the principal limiting factor for the application of bioremediation.

One of the promising directions of oil-polluted aquatic and soil ecosystem remediation is the use of combined methods based on oil-degrading microorganisms and various sorbents. Previously, laboratory and model field studies using mineral (thermally activated vermiculite), organic (peat), and carbonaceous (activated carbon and biochar) sorbents have shown the effectiveness and potential of sorption–biological treatment of soils polluted with oil products at a low temperature [28,29,30,31,32,33]. The use of activated carbon reduced phytotoxicity and even stimulated the growth of test plants [34,35].

The aim of this study was to evaluate the efficiency of a sorption–biological method for treatment of oil-polluted coastal substrates of the Barents Sea under low temperatures using the active hydrocarbon-oxidizing bacterial strain *Rhodococcus erythropolis* HO-KS22.

## 2. Materials and Methods

### 2.1. Research Objects

The research objects were soil and sand from the coast of Pechenga Bay of the Barents Sea (69°34′25.4″ N, 31°13′44.6″ E). Substrates were collected in the summer. The soil was sampled in the supralittoral zone (after removing plants) and sand in the littoral zone during low tide with a scoop to a depth of 5 cm at several points over an area of 100 m^2^. In the laboratory, sand and soil samples were sieved through a 2 mm mesh to remove large inclusions. The hydrocarbon content in pure substrates was 41 mg·kg^−1^ for sand and 85 mg·kg^−1^ for soil. The pH value was 6.25 in sand and 3.85 in soil. Dehydrogenase activity was very low in both sand (0.01 mg TPP·10 g^−1^) and soil (0.17 mg TPP·10 g^−1^) (Table 1).

### 2.2. Water–Oil Emulsion

The water–oil emulsion prepared from three types of Prirazlomnoye oil of different densities was used as a pollutant: light (855 g·L^−1^) medium (910 g·L^−1^), and heavy (970 g·L^−1^). The emulsion was prepared over 24 h at a temperature of 10 °C using a mixing device in the Laboratory for Oil Property Studies at the Murmansk Center for Standardization and Metrology (Murmansk, Russia). The oil-to-water ratio was 1:10. Seawater was collected from the northern part of Kola Bay. Volatile and labile components were first removed from the oil by heating it in a water bath to 100 °C for 2 h. This treatment simulated the weathering and photooxidation processes that occur with oil on the surface of the water after a spill.

### 2.3. Sorbents

The following sorbents were used for bacterial localization:-Granular activated carbon (GAC VSK, Nizhny Novgorod, Russia) with granule sizes of 2–3 mm consists of 87–97% carbon and exhibits a sorption capacity for hydrocarbons up to 980 mg·g^−1^ [36].-Thermally activated vermiculite (JSC “Mica Factory”, St. Petersburg, Russia) is a mineral of the aluminosilicate group with a layered structure. When heated, vermiculite swells and increases in volume several fold; it has a high absorption coefficient, with a sorption capacity for hydrocarbons up to 5400 mg·g^−1^ [37].-Highmoor milled peat with a low degree of decomposition (not exceeding 35%), with a hydrocarbon sorption capacity ranging from 1560 to 1825 mg·g^−1^ [38].

### 2.4. Hydrocarbon-Oxidizing Strain Rhodococcus Erythropolis HO-KS22

Strain HO-KS22 was isolated from oil well 1010 of the Vostochno-Anzirskoe oil field (Naberezhnye Chelny, Russia). Analysis of the 16S rRNA gene classified the strain as *Rhodococcus erythropolis* (99.8% similarity to the type strain *R. erythropolis* DSM 43066T). The 16S rRNA gene sequence of strain HO-KS22 was deposited in GenBank under accession number MN622878. Strain *Rhodococcus erythropolis* HO-KS22 was deposited in the All-Russian Collection of Microorganisms under the number Ac-2807D. The microorganism is an aerobic organotrophic bacterium, rod-shaped in young cultures and coccoid-rod-shaped in older cultures, Gram-positive, and non-spore-forming. Strain HO-KS22 grows within a temperature range of 5–37 °C (optimum ~28 °C) and NaCl concentrations from 0 to 120 g·L^−1^ (optimum ~10 g). The pH range for growth is 5.5 to 9.5 (optimum pH 7.0–7.5) [39]. The microorganism can utilize a wide range of substrates, including sugars, volatile acids, alcohols, hydrocarbons, and crude oil. Growth on hydrocarbons is accompanied by surfactant formation, leading to decreased surface tension of the medium and interfacial tension at the culture liquid/hexadecane interface [40]. The strain’s ability to actively grow on petroleum and its high urease activity make it suitable for enhanced petroleum recovery and bioremediation applications [41].

### 2.5. Experimental Design

The water–oil emulsion of different oil types was added to the substrates (soil and sand) at 1% by weight and thoroughly mixed. Polluted substrates (200 g) were amended with sorbents (peat, GAC, vermiculite), nitrogen–phosphorus–potassium fertilizer (NPK 1:1:1), and a suspension of hydrocarbon-oxidizing bacteria. Substrates and sorbents were not sterilized before use. Sorbents were added at the following proportions relative to substrate weight: peat—5%, GAC—1%, vermiculite—0.5%. Mineral fertilizer was applied at 360 mg per 200 g of substrate, based on commonly accepted agricultural application rates of 60–90 kg·ha^−1^ of nitrogen, phosphorus, and potassium [42]. A bacterial suspension of strain HO-KS22 with a 10^7^ cells·mL^−1^ was added at 4 mL per 200 g of substrate. Control samples (natural attenuation) are the polluted soil and sand with water–oil emulsion without bacterial suspension of strain HO-KS22, sorbents, and mineral fertilizer. We used control treatments to account for abiotic losses and biodegradation of hydrocarbons by the indigenous microbial community (Table 2). All variants were repeated in triplicate.

The soil and sand pots were open. The substrates were moistened with seawater (non-sterile) to maintain humidity in the range of 30–40% throughout the experiment and stirred twice a month. The experiment was at a constant temperature of +10 °C for 3 months. This temperature corresponds to the average air temperature during the growing season on the southwestern coast of the Barents Sea.

To assess the efficiency of hydrocarbon sorption on the sorbents, samples of polluted soil and sand after 3 months of treatment by the sorption–biological method were filled with seawater and stirred on a laboratory shaker for 1 h. The ratio of substrate to seawater was 1:5.

### 2.6. Total Petroleum Hydrocarbon Content

The total petroleum hydrocarbon (TPH) content was determined monthly by infrared (IR) spectrometry using an AN-2 petroleum product analyzer (“Neftekhimavtomatika”, St. Petersburg, Russia), based on measurement of the integral absorption intensity of C–H bonds in methyl and methylene groups within the infrared region. TPH was defined as a complex mixture comprising chain and cyclic hydrocarbons, heteroatomic compounds, resins, and asphaltenes. Additionally, besides hydrocarbons typical of oil and petroleum products, the extract contained components from soil humus and plant residues. To remove these impurities, elution on Al_2_O_3_ was applied; however, this process retained not only soil components but also high-molecular-weight petroleum fractions and polycyclic aromatic hydrocarbons on the sorbent [43]. To quantify high-molecular-weight soil organic components and high-molecular-weight petroleum fractions in substrate samples, the integral absorption intensity of C–H bonds in the extract was measured before elution on Al_2_O_3_. The hydrocarbon content was calculated for dry soil.

### 2.7. The Number of Hydrocarbon-Oxidizing Bacteria

Approximately 10 g of substrate sample was collected from each pot for microbiological analyses following the standard soil sampling protocol. The analyses of the number of cultivable bacteria were started the next day after soil sampling. The number of cultivable hydrocarbon-oxidizing bacteria (HOB) was determined using standard plating methods [44] on specific media with the following composition (g·L^−1^): K_2_HPO_4_·3H_2_O—1, NH_4_Cl—2, MgSO_4_·7H_2_O—0.5, NaCl—0.5, CaCO_3_—1, FeSO_4_·7H_2_O—trace amounts, oil—1 vol. %. The pH of all nutrient media was adjusted to 5.0 with either HCl or NaOH. Incubation was carried out at 25 °C for 7–10 days. The number of HOB was calculated for dry soil.

### 2.8. Assessment of Dehydrogenase Activity

Dehydrogenase activity was measured colorimetrically at a wavelength of 540 nm and a cuvette path length of 5 mm. The method is based on the anaerobic reduction of colorless C_19_H_15_ClN_4_ to red C_19_H_16_N_4_ [45].

### 2.9. Measuring pH Value

The hydrogen ion concentration (pH) was determined potentiometrically using a combined electrode and an ion meter in an aqueous suspension of soil or sand (substrate-to-water ratio of 1:4) [46].

### 2.10. Statistical Processing

The data were processed according to standard procedures of descriptive statistics. An ANOVA with a post hoc test was used to compare the content of TPH and other parameters under different treatments. The relationships were analyzed using Pearson correlation. The data were analyzed and visualized using the Microsoft Excel software package (Microsoft 365, version 2508) and R (RStudio, version 2023.3.0.386).

## 3. Results

### 3.1. Content of Total Petroleum Hydrocarbons

In sand, the initial content of petroleum hydrocarbons one day after the pollution of the different oil types ranged from 1894 to 3967 mg·kg^−1^. The amount of hydrocarbon losses due to a combination of abiotic losses and the indigenous microbial community biodegradation over three months ranged from 947 to 2478 mg·kg^−1^ (Figure 1A–C; Appendix A), and the hydrocarbon content decreased by 54–64% in sand. The processes of destruction in the samples with oil pollution are going mainly by physical and chemical environmental factors, not by microbial oxidation, due to a low intensity of natural attenuation processes in the soils of the Arctic zone under the action of indigenous microorganisms [47].

The maximum rate of hydrocarbon degradation in sand without treatment was observed three months after pollution, with the rate gradually increasing throughout the experiment. The average rate of hydrocarbon degradation of sand polluted with light oil (28 mg·d^−1^) was twice as high as that in samples polluted with medium and heavy oil (11–16 mg·d^−1^) (Figure 1D–F).

Bioremediation of polluted sand increased the degree of treatment by 7–20%; however, no significant differences were found with different sorbents. The hydrocarbon degradation rate in sand with light oil increased 3–4 times during the first month (from 15 mg·d^−1^ in the control to 46–63 mg·d^−1^) (Figure 1D). The total amount of hydrocarbons degraded during three months of biological treatment of polluted sand was 2905–2972 mg·kg^−1^ for light oil, 1316–1405 mg·kg^−1^ for medium oil, and 1602–1715 mg·kg^−1^ for heavy oil.

The results of ANOVA showed a significant difference only between the natural attenuation and peat treatment after 30 days (*p* < 0.043) for sand polluted with light oil. The treatments of sand polluted with medium and heavy oil were not statistically different (*p* > 0.05).

In soil, the initial content of petroleum hydrocarbons one day after the pollution of the different oil types ranged from 6609 to 7903 mg·kg^−1^. The amount of hydrocarbon losses due to a combination of abiotic losses and the indigenous microbial community biodegradation over three months ranged from 3572 to 5652 mg·kg^−1^ (Figure 2A–C; Appendix A), and the hydrocarbon content in the control samples decreased by 51–72%.

The hydrocarbon degradation rate in soil peaked during the first month after pollution—63 mg·d^−1^ in the variant with medium oil and 40–47 mg·d^−1^ in those with light and heavy oil (Figure 2D–F).

Bioremediation of polluted soil was ineffective (Figure 2A–C). The hydrocarbon content decreased by 51–73% after three months, which is comparable to the results of the variant without treatment. Bioremediation slightly increased the rate of hydrocarbon degradation (Figure 2D–F), but over a longer period the biodegradation rate was comparable or even lower than in the control variant. The amount of hydrocarbons degraded over three months during biological treatment of soil was 5517–5775 mg·kg^−1^ for medium, 3947–4136 mg·kg^−1^ for light, and 3531–3869 mg·kg^−1^ for heavy oil. Hydrocarbons of medium oil were degraded to a greater extent than those of light and heavy oil.

The post hoc Tukey’s test revealed no significant differences between treatments of soil throughout the experiment.

The rate constant increases from 0.009–0.011 without treatment to 0.013–0.016 with sorption–biological treatment in sand (Table 3). The degradation rate constant in the soil remained virtually unchanged during treatment.

### 3.2. Content of High-Molecular Organic Compounds

Petroleum hydrocarbons predominated among the high-molecular organic compounds (HMOCs) extracted from polluted substrates with CCl_4_. The proportion of HMOCs increased up to 37% with increasing density of the oil used in the experiment (Figure 3; Appendix A).

In polluted sand during natural attenuation, the content of HMOCs increased. The most intensive accumulation of HMOCs occurred in the second and third months and was most pronounced in the variant with light oil, where the maximum rate of hydrocarbon destruction was observed (Figure 4A). A strong, but not reliable, negative correlation was established between the content of hydrocarbons and high-molecular compounds in the sand (r = 0.581–0.934).

Sorption–biological treatment resulted in a decrease in the amount of extracted HMOCs from sand. The maximum amount of these components was noted for sand with peat, which is due to the organic nature of the sorbent itself, while the minimum amount was noted for samples with activated carbon and vermiculite, possibly indicating sorption of HMOCs on these sorbents (Figure 4A–C). After three months, the ratio of hydrocarbons to HMOCs in the sand shifted, with an increase in the share of the latter, indicating transformation of petroleum hydrocarbons as a result of biological and chemical degradation (Appendix A).

In the soil a decrease in the content of high-molecular components by 29–60% was noted (Appendix A). A strong positive correlation was established between the content of hydrocarbons and HMOCs, statistically significant for the variants of soil with medium and heavy oil (r = 0.958–0.999, *p* = 0.05). The decrease in the content of HMOCs in the soil can be explained by more complete oxidation of hydrocarbons and incorporation of their transformation products into the organic matter. In sand with low sorption capacity, the products of hydrocarbon transformation, including HMOCs, are more mobile and are easily extracted.

Biological treatment also decreased the content of HMOCs in soil (Figure 4D–F). In this case, the nature of the sorbents and their sorption properties were less pronounced due to the presence of the soil’s organic matrix and its high sorption capacity. In soil, as a result of bioremediation, the ratio of hydrocarbons to HMOCs did not change significantly due to the low degree of their extraction from the soil.

At the same time, the ANOVA test did not reveal significant differences between treatment variants in both sand and soil polluted with different types of oil (*p* > 0.05).

### 3.3. Number of Hydrocarbon-Oxidizing Bacteria

Sand pollution increased the number of hydrocarbon-oxidizing bacteria (HOB), for which hydrocarbons are a nutrient and energy substrate. In the natural attenuation the number of HOB increased by 300 times after one month. And, despite a decrease in the number after two months, it remained significantly higher than the initial level by 47–67 times.

Sorption–biological treatment of sand with the HO-KS22 strain and peat resulted in an increase in the number of HOB from 0.6 × 10^6^ to 3.8 × 10^9^ cells·g^−1^ after one month. This is due to the large amount of easily oxidized hydrocarbons in light oil, as well as the presence of bacteria in peat capable of utilizing hydrocarbons (Figure 5A–C). However, after two months, the number of HOB in this variant decreased, probably due to a decrease in the amount of hydrocarbons available for oxidation. The use of activated carbon and vermiculite also increased the amount of HOB after one month to 0.45 × 10^9^ and 0.23 × 10^9^ cells·g^−1^, respectively. After three months, the number of this group of bacteria was 3–7 times higher than in natural attenuation.

In sand with medium and heavy oil, the addition of peat decreased HOB number during the first month compared to the natural attenuation (Figure 5B,C). The addition of extra organic matter to the sand created a more accessible nutrient substrate than the hydrocarbons of medium and heavy oil, which may have slowed the growth of the HOB. After two months, the HOB number exceeded the control values by 8–9 times and reached its maximum.

However, the ANOVA results did not reveal any significant effect of different treatments on the amount of HOB in polluted sand.

Pollution of soil also increased the HOB number. Subsequently, the number of HOB significantly decreased and, after three months, did not exceed the initial values.

Sorption–biological treatment of soil with light oil did not increase the HOB number. Maximum HOB number was after one month in soil with medium oil, and after two months with heavy oil (Figure 5D–F). Unlike the sand, the maximum HOB number during soil bioremediation was with the activated carbon and vermiculite, but not peat.

The maximum number of HOB in the sand did not exceed 5 × 10^8^ cells·g^−1^ in all variants, except for the samples with light oil and addition of peat. The number of bacteria in the soil reached 3 × 10^9^ cells·g^−1^, and the lowest values were noted with heavy oil (2–6 × 10^8^ cells·g^−1^). However, despite the increase in the number of HOB, bioremediation of soil did not significantly affect the petroleum hydrocarbon content. The native microbial communities present in the soil transformed hydrocarbons without the introduction of an active strain of HOB. The results of ANOVA also revealed no significant effect of different treatment variants on the amount of HOB in polluted soil.

### 3.4. Dehydrogenase Activity

Soil dehydrogenase activity is an important indicator of biological activity and the functioning of the soil microbiota. Dehydrogenase is a marker of the soil’s ability to recover from anthropogenic impacts.

Dehydrogenase activity in the natural attenuation sand samples increased 2–4 times one month after pollution but still remained very low (0.02–0.04 mg TPP·10 g^−1^). This short-term increase in enzyme activity was directly related to the pollution of the sand with water–oil emulsion, i.e., to the influx of additional organic matter. A strong positive correlation was established between the number of HOB and dehydrogenase activity in sand, significant for the variants with light and medium oil (r = 0.963–0.973, *p* = 0.05).

Sorption–biological treatment of sand increased in dehydrogenase activity in all variants, the most with light oil. After two months, the dehydrogenase activity decreased with medium and heavy oil or remained slightly elevated with light oil. Among the sorbents, the granulated activated carbon significantly increased dehydrogenase activity—from 3 to 63 times—which corresponds to high activity (Appendix A–C).

A Tukey test showed significant differences between GAC and natural attenuation as well as between GAC and vermiculite after 30 days in sand polluted with light oil (*p* = 0.045–0.049).

The dehydrogenase activity in the soil samples after pollution was 0.18–0.21 mg TPP·10 g^−1^. After one month, in the natural attenuation samples with light oil, the enzyme activity increased by 1.4–2.4 times, which corresponds to low activity, and after three months, it decreased to the initial very low values (0.15–0.16 mg TPP·10 g^−1^).

Sorption–biological treatment of soil increased dehydrogenase activity during the first month in all variants by 1.5–2.5 times, but by the end of the third month the activity decreased again. Correlation analysis revealed a positive relationship between dehydrogenase activity and the rate of hydrocarbon degradation in soil (r = 0.411–0.729), significant for the variants with medium and heavy oil (r = 0.675–0.729, *p* = 0.05). The highest enzyme activity in soil, as well as the maximum rate of hydrocarbon degradation, occurred during the first month of the experiment. Furthermore, the highest dehydrogenase activity during treatment of soil was in the variants with medium oil, which also demonstrated the most intensive degradation of hydrocarbons.

A Tukey test showed significant differences between peat and natural attenuation as well as between peat and vermiculite after 30–60 days in soil polluted with medium oil (*p* = 0.004–0.037) and between peat and natural attenuation with heavy oil (*p* = 0.022).

### 3.5. pH Value

In the natural attenuation, two months after pollution, a decrease in the pH value of sand and soil by 0.4–0.9 units was observed. At the same time, the change in pH was more pronounced in sand due to soil already being acid.

Sorption–biological treatment did not significantly affect the pH value of the sand. As in the control, a decrease in pH was noted, especially when peat was used (Appendix A–C). Bioremediation of soil reduced the acidification—after three months, the soil pH was in the range of 3.9–4.3, which was close to the background values, whereas in the natural attenuation variant, the pH decreased to 3.2–3.5 (Appendix A–F).

An ANOVA test showed significant differences between vermiculite and natural attenuation after 30 days in soil polluted with light oil (*p* = 0.008) and between GAC and natural attenuation as well as between vermiculite and natural attenuation after 30–60 days with medium oil (*p* = 0.003–0.006). For soil polluted with heavy oil, all treatments were significantly different from natural attenuation after 60–90 days (*p* = 0.006–0.041).

### 3.6. Desorption of Hydrocarbons from Polluted Soil and Sand

The hydrocarbon content in pure seawater did not exceed 0.025 mg·L^−1^. No more than 0.020 mg of organic matter, defined as hydrocarbons, passed into water from pure sand, which did not exceed the maximum permissible concentration (MPC) for water for fishery (0.050 mg·L^−1^) and domestic purposes (0.3 mg·L^−1^). At the same time, much more hydrocarbons desorbed into water from polluted sand (from 0.119 to 0.429 mg·L^−1^) (Figure 6A). Hydrocarbon desorbed from sand after sorption–biological treatment was close to the values obtained for pure sand (0.010–0.036 mg·L^−1^), with the exception of the variant with heavy oil (0.059–0.084 mg·L^−1^). However, even with heavy oil, hydrocarbon desorption after treatment decreased by 3.1–5.4 times, which confirms the effectiveness of the sorption–biological method.

From 0.068 to 0.121 mg·L^−1^ desorbed into water from polluted soil, which is less than from polluted sand (Figure 6B). This is a consequence of the higher sorption capacity of the soil, the particles of which retain hydrocarbons more firmly. As in the sand, the hydrocarbon desorption from polluted soil after sorption–biological treatment decreased. For the variants with light and medium oil, the obtained results were comparable with the values for pure soil or even lower (0.025–0.063 mg·L^−1^).

Higher sorption of hydrocarbons was typical for the variants with thermally activated vermiculite and activated carbon. Thus, the use of these sorbents prevented hydrocarbon desorption in seawater and secondary pollution of ecosystems, including previously unpolluted ones.

## 4. Discussion

The results of hydrocarbon degradation rates obtained in our study are in many ways typical for the Arctic. The rate of hydrocarbon degradation in polluted soil (Murmansk region) was 24–105 mg·d^−1^ [31]. The hydrocarbon degradation rate constants we obtained are consistent with the results of oil degradation studies in the Arctic, where k was 0.010–0.014 [48], 0.017 [49,50], and 0.0095–0.011 [51]. During the natural attenuation, the hydrocarbon content in polluted substrates decreased by 51–72%, indicating a high rate of degradation as a result of the action of abiotic factors and indigenous hydrocarbon-degrading microorganisms.

The results demonstrated a difference in the dynamics of oil hydrocarbon degradation due to natural attenuation in different coastal substrates: in soil, this process was most intensive during the first month after pollution, while in sand during the third month. The reason for the different degradation of hydrocarbons in sand and soil was the physical and chemical differences in the substrates. Physical composition of sediments can impact hydrocarbon biodegradation. Microscopic oil–particle aggregates with fine sediments may facilitate bioremediation. But macroscopic oil–sediment aggregates may negatively impact to bioremediation [3,52]. The high rate of petroleum hydrocarbon degradation in soil is attributed to more favorable conditions (availability of nutrients, buffering capacity, indigenous hydrocarbon-degrading microorganisms). The importance of nitrogen, phosphorus, and iron on shorelines has been widely characterized as these nutrients are often limited in sediments [53,54]. The lower rate of hydrocarbon degradation in sand in the first month may be due to the toxicity of hydrocarbons. A high concentration of hydrocarbons reduces the biodegradation rate by creating a toxic environment for microorganisms. Low molecular-weight alkanes and aromatic compounds show acute toxicity, and polyaromatic hydrocarbons cause chronic toxicity for microorganisms. Low Arctic temperatures contribute to increasing soil toxicity levels by reducing the evaporation rate of hydrocarbon compounds [11].

Light and medium oil were more biodegradable than heavy oil, owing to the higher proportion of high-molecular compounds in the latter, which are less available to microorganisms. Hydrocarbons of heavy oil were less susceptible to microbiological degradation and also decreased with the number of HOB and dehydrogenase activity in soil. A crude oil concentration of 1% can be toxic and reduces the abundance of hydrocarbon-oxidizing microorganisms [55]. At low temperatures, biodegradation of high-viscosity oil was delayed and volatilization of toxic short-chain alkanes decreased [56]. Given the behavior of dense and viscous oil, a ban on the use of heavy fuel oil (HFO) in the Arctic was introduced, aimed at reducing the negative impact caused by the use and spillage of HFO on the Arctic environment [1].

The use of hydrocarbon-oxidizing bacteria *Rhodococcus erythropolis* HO-KS22 with peat increased the rate of oil hydrocarbon degradation in coastal sand with light oil by 3–4 times during the first month and accelerated treatment by 20% over the three months. This result further confirms the fact that bioremediation appears to improve the rate, but not the extent, of microbial degradation of hydrocarbons [3]. It has been shown that the addition of nutrients for biostimulation has limited effectiveness in accelerating the biodegradation process, despite its initial promotion of aliphatic hydrocarbons within a constrained timeframe [57]. The acceleration of the hydrocarbon degradation rate in polluted sand in the first month after treatment is due to the introduction of fertilizers, sorbents, and hydrocarbon-oxidizing bacteria into the initially poor substrate. During the first month an increase in dehydrogenase activity was observed in all variants which is due to the increase in the number of hydrocarbon-oxidizing bacteria. Bioremediation reduced the amount of extractable high-molecular organic compounds from sand, which may indicate their sorption onto the sorbents. Among the sorbents used, the granulated activated carbon increased in dehydrogenase activity better, while the peat contributed to the increase in the HOB number due to the native microflora.

Bioremediation of coastal soil was ineffective, and the residual oil hydrocarbon content did not differ significantly from the natural attenuation. The low efficiency of the applied strain may be explained by both the relatively high abundance of native HOB in the soil and specific soil conditions, such as low pH values and temperature, which fall outside the optimal range for this strain. However, this approach increased dehydrogenase activity. Additionally, it mitigated acidification of the soil during the remediation process, primarily due to the use of sorbents.

The ANOVA results did not reveal any significant effect of different treatments on the amount of HOB in polluted sand and soil. No significant correlations were found between the number of HOB and the rate of microbiological oxidation of petroleum hydrocarbons. The gradual decrease in the number of hydrocarbon-oxidizing bacteria under natural attenuation is probably because most of the easily degradable TPH was already degraded [5]. The reduction in bacterial numbers and the slowing of the degradation rate three months after treatment are likely due to reduced availability of TPH for microbial degradation, nutrient limitation, or accumulation of toxic transformation products [58]. Changes in the number of hydrocarbon-oxidizing bacteria under treatment may be a consequence of the instability of the microbial community at this stage. Oil spills can reduce microbial diversity when the spill occurs but can increase the abundance of hydrocarbon-degrading microorganisms [55] and microbial community composition of the treated polluted substrate can quickly change compared to untreated substrates [59].

The reasons for the low efficiency of the strain HO-KS22 could be the negative environmental factors (low temperatures and pH values) that are beyond the optimal limits for this strain, inability to degrade high-molecular-weight hydrocarbons present in oil, competitive interactions with the indigenous microflora of the substrates, as well as a deficiency of nutrients, which confirms the difficulty of biological treatment of shorelines in low temperatures and the multicomponent composition of oil.

A definite positive aspect was the reduction in hydrocarbon desorption from polluted substrate in seawater with the use of sorbents. In the process of cleaning coastal areas, this will help avoid unwanted pollution and reduce the impact on ecosystems.

The climatic and geographical features of the Arctic Sea coasts and the Barents Sea in particular make remediation of polluted areas difficult. The diversity of shore types and substrates requires an individual cleanup plan that takes into account nutrient content, microbiological activity, pH value, physical particle size, and buffering capacity of soils. Available hydrocarbon-degrading strains of microorganisms are not always effective in northern latitudes. Additionally, the results of the study of the sorbents’ effectiveness for oil, obtained under standard laboratory conditions, differ from real field conditions due to natural and anthropogenic factors that are not taken into account by manufacturers of sorption materials [60].

## 5. Conclusions

The strain *Rhodococcus erythropolis* HO-KS22, which has demonstrated the ability to use hydrocarbons in laboratory conditions, can be used to clean coastal sand of light oil, accelerating this process by 20%, especially in the first month after pollution. However, for the successful biological remediation of oil-polluted soils of the Barents Sea coast, further selection of active hydrocarbon-oxidizing strains adapted to the low pH values and temperatures of this region is required. Heavy oil containing a large number of high-molecular compounds is difficult to oxidize biologically, even with the use of the active hydrocarbon-oxidizing strain *Rhodococcus erythropolis* HO-KS22. Cleaning and restoration of coastal soils by sorption–biological methods may be ineffective.

The use of sorbents both in sand and in soil helps prevent secondary pollution of coastal ecosystems, since it significantly reduces hydrocarbon desorption and their leaching by water.

Among the studied sorbents, it is advisable to use granulated activated carbon or thermally activated vermiculite for sand cleaning, since they reduce the content of high-molecular compounds, including potentially toxic hydrocarbon transformation products, maintain a high number of hydrocarbon-oxidizing bacteria, and increase the activity of dehydrogenase. The use of peat, despite the fact that it contains a large number of bacteria, can cause excessive acidification of the substrate, which was shown in our work, and also change the structure of the sand.

The introduction of sorbents into polluted soil, although it does not affect the degree of treatment of hydrocarbons, also helps to avoid excessive acidification after the introduction of mineral fertilizers and the accumulation of hydrocarbon transformation products.

The application of hydrocarbon-degrading microbial preparations without consideration of regional soil and climatic conditions may be ineffective and increase the cost of remediation efforts without significantly improving the polluted sites.

## Figures and Tables

**Figure 1 microorganisms-13-02181-f001:**
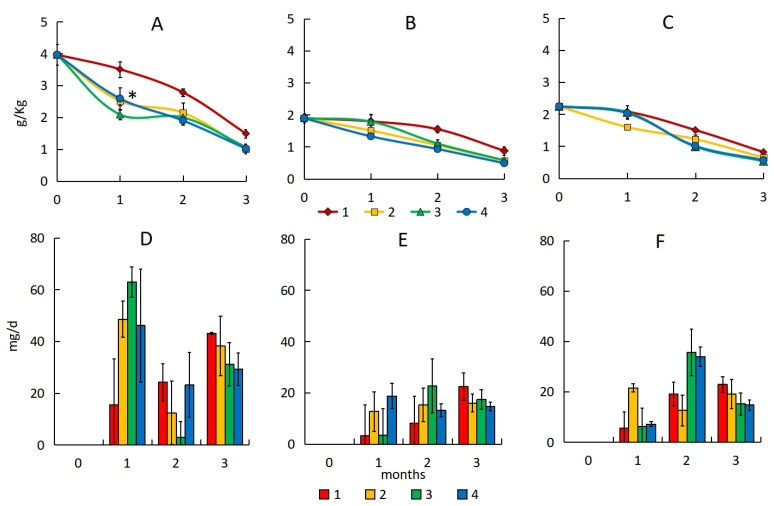
The petroleum hydrocarbon content with emulsion of light (**A**), medium (**B**), and heavy (**C**) oil and hydrocarbon degradation rate with emulsion of light (**D**), medium (**E**), and heavy (**F**) oil in sand: 1—natural attenuation, 2—*R. erythropolis* and peat, 3—*R. erythropolis* and activated carbon, 4—*R. erythropolis* and vermiculite. *—reliable differences between the natural attenuation and the sorption–biological treatment with a significance level of 0.05.

**Figure 2 microorganisms-13-02181-f002:**
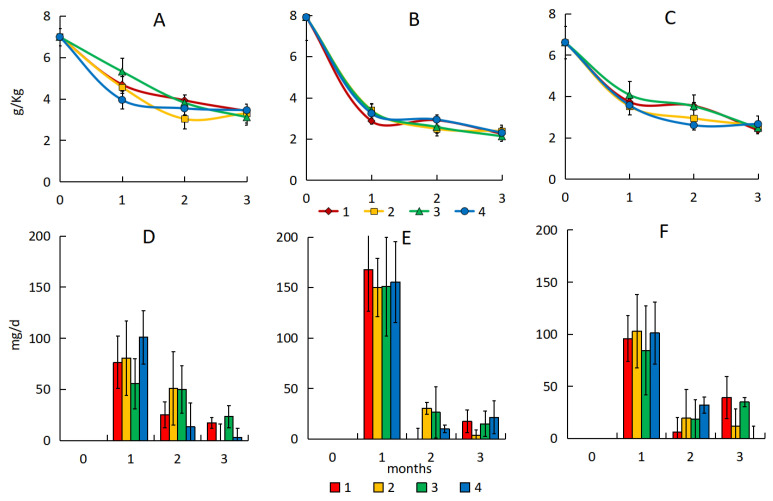
The petroleum hydrocarbon content with emulsion of light (**A**), medium (**B**), and heavy (**C**) oil and hydrocarbon degradation rate with emulsion of light (**D**), medium (**E**), and heavy (**F**) oil in soil. 1—natural attenuation, 2—*R. erythropolis* and peat, 3—*R. erythropolis* and activated carbon, 4—*R. erythropolis* and vermiculite.

**Figure 3 microorganisms-13-02181-f003:**
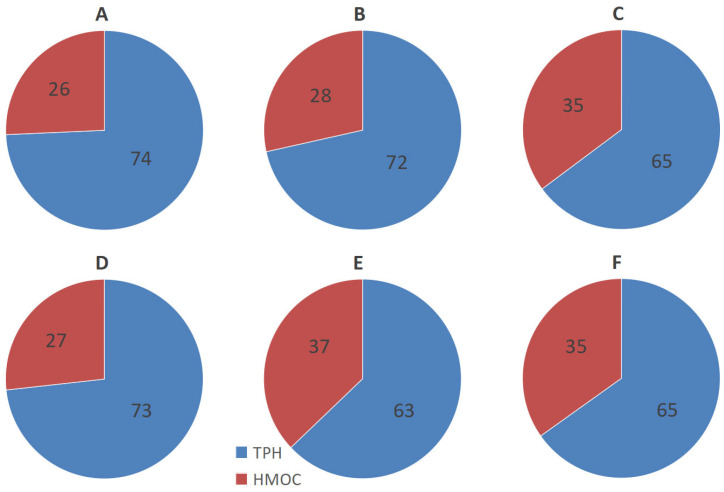
The ratio of TPH to HMOCs in the sand with emulsion of light (**A**), medium (**B**), and heavy (**C**) oil and in the soil with emulsion of light (**D**), medium (**E**), and heavy (**F**) oil before treatment.

**Figure 4 microorganisms-13-02181-f004:**
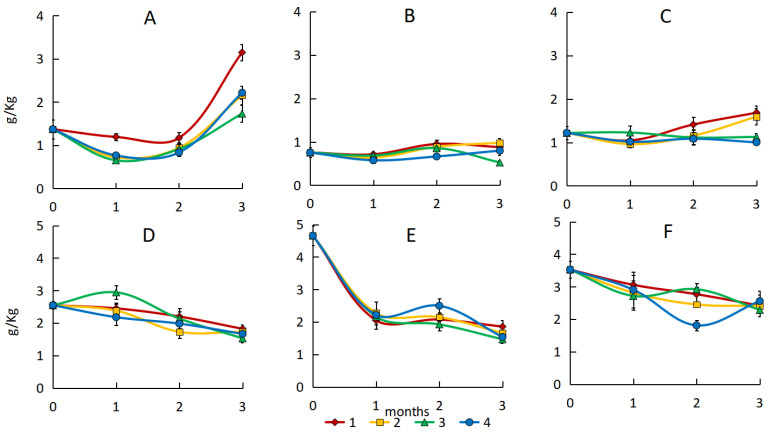
The high-molecular organic compound content in the sand with emulsion of light (**A**), medium (**B**), and heavy (**C**) oil and in the soil with emulsion of light (**D**), medium (**E**), and heavy (**F**) oil. 1—natural attenuation, 2—*R. erythropolis* and peat, 3—*R. erythropolis* and activated carbon, 4—*R. erythropolis* and vermiculite.

**Figure 5 microorganisms-13-02181-f005:**
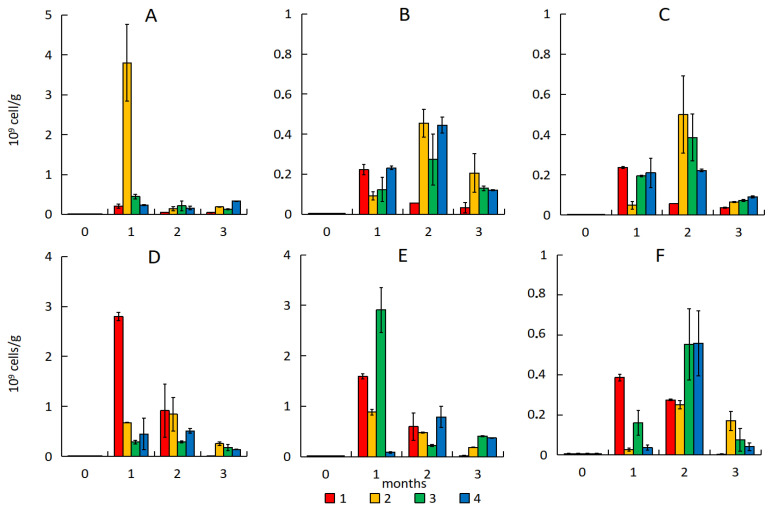
The number of HOB in the sand with emulsion of light (**A**), medium (**B**), heavy (**C**) oil and in the soil with emulsion of light (**D**), medium (**E**), and heavy (**F**) oil. 1—natural attenuation, 2—*R. erythropolis* and peat, 3—*R. erythropolis* and activated carbon, 4—*R. erythropolis* and vermiculite.

**Figure 6 microorganisms-13-02181-f006:**
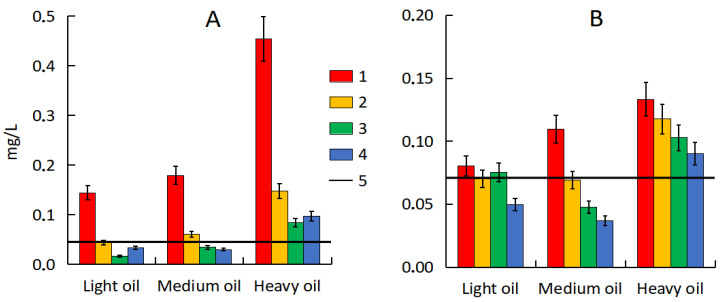
Desorption of hydrocarbons from polluted sand (**A**) and soil (**B**). 1—natural attenuation, 2—*R. erythropolis* and peat, 3—*R. erythropolis* and activated carbon, 4—*R. erythropolis* and vermiculite, 5—background.

**Table 1 microorganisms-13-02181-t001:** Characteristics of the substrates for the laboratory experiment.

Substrates	TOC, %	pH	Dehydrogenase Activity, mg TPP·10 g^−1^	TPH, mg·kg^−1^	Description of the Site
Sand	0.10	6.25	0.01	41	Sandy beach
Soil	4.74	3.85	0.17	85	Meadow shores: 0–2 cm—organogenic, deeper than 2 cm—loam

**Table 2 microorganisms-13-02181-t002:** Design of the laboratory experiment.

Variants *	Sorbents, g			Mineral Fertilizers, g	Bacterial Suspension, mL
	Peat	GAC	VER		
1 (Natural attenuation)	-	-	-	-	-
2 (HO-KS22 + Peat)	10	-	-	0.36	4
3 (HO-KS22 + GAC)	-	2	-	0.36	4
4 (HO-KS22 + vermiculite)	-	-	1	0.36	4

* Each variant was applied to soil and sand and three types of oil.

**Table 3 microorganisms-13-02181-t003:** The degradation rate constants (90 days).

	Sand			Soil		
	Light Oil	Medium Oil	Heavy Oil	Light Oil	Medium Oil	Heavy Oil
Natural attenuation	0.011	0.009	0.011	0.008	0.014	0.011
*R. erythropolis* and peat	0.015	0.014	0.014	0.008	0.013	0.010
*R. erythropolis* and activated carbon	0.015	0.013	0.016	0.009	0.015	0.011
*R. erythropolis* and vermiculite	0.015	0.015	0.015	0.008	0.014	0.010

## Data Availability

The original contributions presented in this study are included in the article/Appendix A. Further inquiries can be directed to the corresponding author.

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
