# Peer review of "Sorption–Biological Treatment of Coastal Substrates of the Barents Sea in Low Temperature Using the *Rhodococcus erythropolis* Strain HO-KS22"

_microorganisms, 2025, doi:10.3390/microorganisms13092181_

Round 1
Reviewer 1 Report
Comments and Suggestions for Authors
Dear authors,
The work is on a relevant topic because the authors have associated bioaugmentation and environmental friendly physical treatment; however the report demands more than 26 references. The positive side of the study is to assess microbial behavior in such environmental conditions during hydrocarbon degradation. The negative side regards the lack of references as well a deep discussion of results. Some important modifications must be made to enhance the quality of presentation. Here are my suggestions and corrections:
- Introduction: in this section, it needs to include references in relevant information such as, the richness of Arctic continental shelf; risk zones related to environmental impacts; natural attenuation which is a monitoring methodology in place of a passive process…
- Page 2, paragraph 5: the correct is to place Actinobacteria (and not actinobacteria) in the part of bacterial mention. Note that it sounds better if you list Actinobacteria’s genera. Based on your classification for fungal cells, the word “fungi” actually refers to Filamentous fungi to be oppositive to yeast.
- Section 2: pay attention of international units and I recommend reviewing the full text: day (d); hour (h), milligrams per liter (mg/L), kilogram (Kg)…
- Section 2.5: were the reactors covered in order to prevent volatilization? Which was the humidity? The soil and sand were watered? Have you considered abiotic losses?
- Section 2.6: use chemical names of compound equally in full text. aluminum oxide or NaCl (as if in section 2.7)
- Section 2.7: please add reference for counting.
- Section 2.9: the same as suggested in section 2.7.
- Section 3.1: the values of reduction of contaminant were calculated by considering abiotic losses? Use at least a theorical value for Arctic zones. All percentages should be revised.
- Page 9: consider integer values for cells, i.e., 0,5x10e9 is 5x10e8
- Section 3.5: the decrease of pH is more pronounced in sand due to soil being already acid. Rephase the results.
- Discussion: briefly, this section is similar to a repetition of results and not a full discussion of the results. There is the need of provide information which means adding new references. This topic is exhaustively discussed and there won’t be difficult to find similar works. 1st paragraph need examples for the “favorable conditions” whose provide a rapid kinetic in the first part of biodegradation studies; 3rd paragraph needs to provide possible reasons for your findings; 4th paragraph is not concise; 5th paragraph needs comparisons.
- It must be included in discussion:
- Abiotic loss in studies set in cold environments
- The fact of first moment of bioremediation is accelerated
- The expected return to initial values of number of cell after the treatment (reasons and facts)
- Possible causes for bioaugmentation failed. Was carried out Rhodococcus counting as well as activity tests with autochthonous microbiota?
Author Response
Dear Reviewer. Thank you very much for taking the time to review this manuscript. Please find the responses below and the corresponding revisions in the re-submitted file. We appreciate your comments and have tried to take them into account to improve our manuscript.
1. Introduction: in this section, it needs to include references in relevant information such as, the richness of Arctic continental shelf; risk zones related to environmental impacts; natural attenuation which is a monitoring methodology in place of a passive process…
We have corrected the Introduction section to reflect your comments.
2. Page 2, paragraph 5: the correct is to place Actinobacteria (and not actinobacteria) in the part of bacterial mention. Note that it sounds better if you list Actinobacteria’s genera. Based on your classification for fungal cells, the word “fungi” actually refers to Filamentous fungi to be oppositive to yeast.
We have changed paragraph 5 to include some Actinobacteria genera.
3. Section 2: pay attention of international units and I recommend reviewing the full text: day (d); hour (h), milligrams per liter (mg/L), kilogram (Kg)…
International units of measurement have been changed all the text
4. Section 2.5: were the reactors covered in order to prevent volatilization? Which was the humidity? The soil and sand were watered? Have you considered abiotic losses?
Section 2.5 has been updated to include information on the experimental conditions.
The soil and sand containers were open. The substrates were moistened with seawater (non-sterile) to maintain humidity in the range of 30-40% throughout the experiment and stirred twice a month. Control samples (natural attenuation) are the soil and sand polluted with water–oil emulsion without bacterial suspension of strain HO-KS22, sorbents and mineral fertilizer. We used control treatments to account for abiotic losses and biodegradation of hydrocarbons by the indigenous microbial community.
5. Section 2.6: use chemical names of compound equally in full text. aluminum oxide or NaCl (as if in section 2.7)
Chemical names of compounds are given as formulas all the text.
6. Section 2.7: please add reference for counting.
The reference to the methodology has been added to the text.
7. Section 2.9: the same as suggested in section 2.7.
The reference to the methodology has been added to the text.
8. Section 3.1: the values of reduction of contaminant were calculated by considering abiotic losses? Use at least a theorical value for Arctic zones. All percentages should be revised.
Using the control variant (natural attenuation), we took into account abiotic losses and biodegradation of hydrocarbons due to the indigenous microbial community. We compared the control values with the results after sorption-biological treatment. We have additionally indicated this in the text to clarify this point.
9. Page 9: consider integer values for cells, i.e., 0,5x10e9 is 5x10e8
We changed the values for the number of cells.
10. Section 3.5: the decrease of pH is more pronounced in sand due to soil being already acid. Rephase the results.
We have rephrased the sentence and made changes to the text.
11. Discussion: briefly, this section is similar to a repetition of results and not a full discussion of the results. There is the need of provide information which means adding new references. This topic is exhaustively discussed and there won’t be difficult to find similar works. 1st paragraph need examples for the “favorable conditions” whose provide a rapid kinetic in the first part of biodegradation studies; 3rd paragraph needs to provide possible reasons for your findings; 4th paragraph is not concise; 5th paragraph needs comparisons.
It must be included in discussion:
Abiotic loss in studies set in cold environments
The fact of first moment of bioremediation is accelerated
The expected return to initial values of number of cell after the treatment (reasons and facts)
Possible causes for bioaugmentation failed. Was carried out Rhodococcus counting as well as activity tests with autochthonous microbiota?
We have taken your comments into account and included the suggested questions in the Discussion.
Reviewer 2 Report
Comments and Suggestions for Authors
The paper presents an interesting study and addresses a relevant topic, especially considering the environmental significance of oil pollution in cold coastal regions. The investigation of the efficiency of the sorption-biological method for treating oil-polluted coastal substrates (soil and sand) of the Barents Sea at low temperature (10 °C) using the active hydrocarbon-oxidizing bacterial strain Rhodococcus erythropolis HO-KS22 is valuable and well-structured.
The results are promising, particularly the observed increase in hydrocarbon degradation rates in sand and soil when treated with the bacterial strain, as well as the positive effects of sorbents in reducing desorption and secondary pollution. The findings highlight the potential of combining biological treatment with sorbents as a remediation strategy following emergency oil spills.
However, certain aspects of the manuscript require revision and clarification before publication. The discussion would benefit from a more detailed explanation of the limitations of the method, particularly concerning the variability of soil and climatic conditions in the Barents Sea region. Further elaboration on the need for bacterial strains resistant to low pH and low temperature would strengthen the argument. Other comments are in the manuscript.

Author Response
Dear Reviewer. Thank you very much for taking the time to review this manuscript. Please find the responses below and the corresponding revisions in the re-submitted file. We appreciate your comments and have tried to take them into account to improve our manuscript.
1. The discussion would benefit from a more detailed explanation of the limitations of the method, particularly concerning the variability of soil and climatic conditions in the Barents Sea region.
We have added information about the Barents Sea region to the Introduction and Discussion.
2. Describe how did You sampled soil?
We have added a description of sampling to the text.
3. How did You determined this values?
A description of the method for determining total petroleum hydrocarbons is given in this section below.
4. Add manufacturer
We have added the manufacturer's name to the text
5. Add a table with initial conditions!
We have added the table with the characteristics of the substrates and the table with the design of the experiment
6. Add initial CFU of suspension and the voume of added suspension
This information is provided in the text and in the table 2
7. Why at this temperature?
This temperature corresponds to the average air temperature during the growing season on the southwestern coast of the Barents Sea.
We have added this information to the text.
8. Per dry or volatile matter? Please indicate.
The hydrocarbon content was calculated based on the dry soil sample. We have added this information to section 2.6.
9. Per dry or volatile matter? Please indicate.
The number of hydrocarbon- oxidizing bacteria was calculated on a dry soil. We have added this information to section 2.7.
Reviewer 3 Report
Comments and Suggestions for Authors
Dear Authors,
Manuscript needs statistics to be added (then the text could be checked in terms of statistically significant differences) and Discussion must be expanded.
Detailed remarks below:
Page 2 – “A relatively large group of hydrocarbon-oxidizing microorganisms is present in the soils. This group includes bacteria of the genera Alcaligenes, Agrobacterium, Arthrobacter, Bacillus, Flavobacterium, Pseudomonas, Rhodococcus; fungi of the genera Aspergillus, Mortierella Penicillium, ……..” -give reference
Page 3 – “After this, the oil acquired characteristics similar to those of oil that had remained on the water surface for two months” – How do you know that?
Page 4 – Paragraph 2.5 – list all the treatments in a clear way – in this form and at this point of the text we do not know if the sorbents were added together or separately. In how many repetitions the experiment was performed?
“The number of hydrocarbon-oxidizing bacteria (HOB) was determined….” – please be more detailed in description of this method
Statistics – why two-sample t-test was used? You have more than two treatments, so more appropriate will be Anova with post-hoc tests like Duncan or LSD. The goal was to compare effectiveness of these four treatments, so you should compare them statistically. You may use different letters for marking statistical differences between treatments at different months. Besides, even results of this two sample t-test are not shown
Page 5 and in the rest of the text– I can not find Suplementary materials in the system, so it is impossible to check parts of the text which are referring to supplementary materials.
Page 8 –
“The number of HOB increased by 300 times.” – in what treatment? In which month? – please be more specific
“Sorption-biological treatment of sand with the HO-KS22 bacterial strain and peat increased the number of HOB even greater.” – greater then what?
Moreover – logically treatment with vermiculate should also be described in the text.
Discussion is mostly repetition of the results and based almost only on your own. It should be much more elaborated. Each analysed parameter should be discussed – the best way is to do that separately (separate paragraphs for content of TPH, then for high-molecular compounds, dehydrogenase, etc.).
Author Response
Dear Reviewer. Thank you very much for taking the time to review this manuscript. Please find the responses below and the corresponding revisions in the re-submitted file. We appreciate your comments and have tried to take them into account to improve our manuscript.
1. Manuscript needs statistics to be added (then the text could be checked in terms of statistically significant differences) and Discussion must be expanded.
In figures 1 and 4 we have indicated reliable differences between the natural attenuation and the sorption-biological treatment with a significance level of 0.05.
The Discussion has been largely corrected and expanded.
2. Page 2 – “A relatively large group of hydrocarbon-oxidizing microorganisms is present in the soils. This group includes bacteria of the genera Alcaligenes, Agrobacterium, Arthrobacter, Bacillus, Flavobacterium, Pseudomonas, Rhodococcus; fungi of the genera Aspergillus, Mortierella Penicillium, ……..” -give reference
We have indicated references in the text
3. Page 3 – “After this, the oil acquired characteristics similar to those of oil that had remained on the water surface for two months” – How do you know that?
We agree with the comment and have formulated the proposal differently
4. Page 4 – Paragraph 2.5 – list all the treatments in a clear way – in this form and at this point of the text we do not know if the sorbents were added together or separately. In how many repetitions the experiment was performed?
We have changed the description of the experiment and added a table with the design of the laboratory experiment
5. “The number of hydrocarbon-oxidizing bacteria (HOB) was determined….” – please be more detailed in description of this method
We have corrected section 2.7 to describe the method in more detail.
6. Statistics – why two-sample t-test was used? You have more than two treatments, so more appropriate will be Anova with post-hoc tests like Duncan or LSD. The goal was to compare effectiveness of these four treatments, so you should compare them statistically. You may use different letters for marking statistical differences between treatments at different months. Besides, even results of this two sample t-test are not shown
We have indicated correlation results in the text where this is significant. Also in Figures 1 and 4 we indicate the differences between natural attenuation and sorption-biological treatment, based on the results of the t-test with a significance level of 0.05.
Unfortunately, we did not conduct the ANOVA test for this study due to insufficient sample size. Preliminary analysis based on the t-test showed no significant differences between the samples. In an new study, we are using sorbents and microorganisms separately to better understand the influence of each factor on hydrocarbon degradation using ANOVA.
7. Page 5 and in the rest of the text– I can not find Suplementary materials in the system, so it is impossible to check parts of the text which are referring to supplementary materials.
We have added additional materials to the text of the article on pages 16-19.
8. Page 8 – “The number of HOB increased by 300 times.” – in what treatment? In which month? – please be more specific
In the natural attenuation the number of HOB increased by 300 times after one month. We have made changes to the text.
9. “Sorption-biological treatment of sand with the HO-KS22 bacterial strain and peat increased the number of HOB even greater.” – greater then what?
We have changed the sentence to make it clearer. Sorption-biological treatment of sand with the HO-KS22 strain and peat resulted in an increase in the number of HOB from 0.6•106 to 3.8•109 cells•g-1 after one month.
10. Moreover – logically treatment with vermiculate should also be described in the text.
We have added a short description to the text.
11. Discussion is mostly repetition of the results and based almost only on your own. It should be much more elaborated. Each analysed parameter should be discussed – the best way is to do that separately (separate paragraphs for content of TPH, then for high-molecular compounds, dehydrogenase, etc.).
The Discussion section has been revised and supplemented to better analyze the data obtained.
Round 2
Reviewer 1 Report
Comments and Suggestions for Authors
Dear authors,
Thank you for accepting the suggestions. The manuscript is well designed and reported.
Author Response
Dear Reviewer, Thank you for your contribution to improving our manuscript and helpful comments. In the attachment you can find the latest version of the manuscript, which takes into account the comments of all reviewers after the second round.

Reviewer 3 Report
Comments and Suggestions for Authors
Dear Authors,
Still, in my opinion t-test is not apropriate for such experiment. It is for comparision of two means. You have four treatments in three repetitions (four means). For such type of experiments Anova should be performed with post-hoc test (or other multiple comparision method with non- parametric test). It should be done for all the parameters, so you know exactly if the treatments you used were working or not. It must be clearly stated in the text if the differences were statisticaly significant or not.
Author Response
Dear Reviewer, Thank you for your valuable comments. We have conducted the ANOVA with post-hoc test of the data. We have included the results of the analysis in the text of the manuscript, indicating significant differences between the variants.
For the content of total petroleum hydrocarbons, the results of ANOVA showed a significant difference only between the Natural attenuation and Peat treatment after 30 days (p < 0.043) for sand polluted with light oil. The treatments of sand polluted with medium and heavy oil were not statistically different (p > 0.05). Also, the post-hoc Tukey's test revealed no significant differences between treatments of soil throughout the experiment.
For the content of high-molecular organic compounds the ANOVA test did not reveal significant differences between treatment variants in both sand and soil polluted with different types of oil (p > 0.05).
The ANOVA results did not reveal any significant effect of different treatments on the amount of HOB in polluted sand and soil.
For the dehydrogenase activity, Tukey test results showed significant differences between GAC and Natural attenuation as well as between GAC and Vermiculite after 30 days in sand polluted with light oil (p = 0.045...0.049), and significant differences between Peat and Natural attenuation as well as between Peat and Vermiculite after 30-60 days in soil polluted with medium oil (p = 0.004...0.037), and between Peat and Natural attenuation with heavy oil (p = 0.022).
For the pH values Tukey test results showed significant differences between Vermiculite and Natural attenuation after 30 days in soil polluted with light oil (p = 0.008), and between GAC and Natural attenuation as well as between Vermiculite and Natural attenuation after 30-60 days with medium oil (p = 0.003...0.006). For soil polluted with heavy oil, all treatments were significantly different from natural attenuation (p = 0.006…0.041).
We also made some changes to the text where necessary based on the results of the ANOVA.